# *Fusarium oxysporum* Associated with Fusarium Wilt on *Pennisetum sinese* in China

**DOI:** 10.3390/pathogens11090999

**Published:** 2022-08-31

**Authors:** Jiaqi Zheng, Liyao Wang, Wenchao Hou, Yuzhu Han

**Affiliations:** 1College of Animal Science and Technology, Southwest University, Chongqing 404100, China; 2Chongqing Key Laboratory of Herbivore Science, Chongqing 404100, China; 3Chongqing Beef Engineering Technology Research Center, Chongqing 404100, China

**Keywords:** *Pennisetum sinese*, *Fusarium oxysporum*, Fusarium wilt

## Abstract

*Pennisetum sinese*, a versatile and adaptable plant, plays an essential role in phytoremediation, soil reclamation, and fodder production. From 2018 to 2021, the occurrence of Fusarium wilt, with symptoms of foliar blight and internal discoloration of the stem, was observed in Chongqing, China. Pathogens were isolated from the symptomatic leaves. Based on morphological characteristics as well as DNA sequences of the 18S ribosomal RNA (SSU), translation elongation factor 1-α (EF1-α), RNA polymerase II subunit 1 (rpb1), and RNA polymerase II second largest subunit (rpb2) genes, the causal agents were identified as *Fusarium oxysporum*. Phylogenetic analysis of the combined dataset of EF1-α, rpb1 and rpb2 showed that pathogenic isolates clustered with *F. oxysporum* strains. The pathogen was reisolated from inoculated and diseased tissues; thus, Koch’s postulates were fulfilled. This is the first report of *F. oxysporum* causing Fusarium wilt on *P. sinese* in China and worldwide.

## 1. Introduction

*Pennisetum sinese* (syn. *Cenchrus flaccidus*), a cross between *Pennisetum purpureum* and *Pennisetum americanum* [1], is extensively distributed in tropical and subtropical regions as a *Gramineae* perennial tall grass [2]. With high biomass production and rich protein and carbohydrates [3], *P. sinese* has been widely cultivated as silage forages, edible fungi culture medium, and shows a valuable potential for biomass energy [4]. Furthermore, *P. sinese* has strong adaptability, which enables its rapid growth in arid environments and contributes to its prominent role in phytoremediation and ecological restoration [5].

As one of the most critical soil-borne pathogens, *Fusarium* species limit plant growth and crop yield [6], and spawn massive agricultural financial losses worldwide [7]. For instance, *Fusarium graminearum*, *F. culmorum*, and *F. avenaceum* can infest *Triticum aestivum* and induce wheat scab. It is commonly found in wheat-producing regions throughout the northwest, southwest, middle, and southeast of China, severely reducing production [8]. *Fusarium* spp. produce carcinogenic mycotoxin that poses a threat to food security and human health [9]. In addition, Fusarium phytotoxicity is regarded as a relevant factor in plant disease severity and progression [10].

*Pennisetum* diseases are known to be induced by fungal species in genera, such as *Curvularia*, *Tolyposporium*, *Puccinia*, *Colletotrichum*, and *Bipolaris*, which lead to the main deterioration factors during cultivation and storage. In China, *P. sinese* production is particularly plagued by leaf blight issues due to being attacked by Gram-negative bacteria members *Pantoea agglomerans* in Chongqing [11] and fungi species *Pyricularia pennisetigena* in Zhanjiang, Guangdong Province [12]. The initial infection manifests as water-soaked brown spots and ultimately the whole leaf blast, consequently affecting the quantity and quality of the herbage.

In May 2018, foliar blight and stem wilt were first observed in the *P. sinese* field in Chongqing City, China, which may potentially harm forage yield. Our study aims to describe this new *P. sinese* wilting disease and identify the causal agent. We confirmed the pathogen associated with *P. sinese* Fusarium wilt through morphological observation, molecular identification, and pathogenicity tests. To our knowledge, this is the first report of Fusarium wilt on *P. sinese* caused by *F. oxysporum* in China and worldwide. Our research will establish the groundwork for primary prevention, diagnosis, and treatment of Fusarium wilt diseases.

## 2. Materials and Methods

### 2.1. Disease Incidence and Sampling Collection

Every May, field surveys were carried out in the 400 m^2^
*P. sinese* fields from 2018 to 2021. The planting base was located at Southwest University (29°22′ N, 105°33′ E), Rongchang District, Chongqing City, China. Typical symptoms of wilting *P. sinese* were recorded from all infected plants. The disease incidence was calculated according to the formula described by Zainudin et al. [13] Seven *P. sinese* fields were randomly selected, and five plots (20 m^2^) were investigated at every site. Approximately 70 plants per plot were included in the analysis. A total of 35 diseased plants were collected and brought to the laboratory.

### 2.2. Fungal Isolation and Morphological Observation

Infected leaf and stem samples were cut into 5–8 mm^2^ pieces, then flashed with running tap water. The symptomatic fragments were immersed in 75% ethanol (*v*/*v*) for 30 s, 1% NaClO (*w*/*v*) for 10 s, and rinsed with sterile distilled water three times. Symptomatic tissues were then transferred onto 9-cm-diameter potato dextrose agar (PDA) and incubated in a biochemical incubator (Boxun BSP100, Shanghai, China) at 25 °C in the dark. Pure cultures of each isolate were obtained by subcultivating hyphal tips on PDA. Fungal colony characteristics were observed on PDA for 5–7 days. After the 10-day-inoculation, the shapes and sizes of the conidia were recorded from at least 100 conidia per isolate under a microscope (Nikon Eclipse E200, Shanghai, China).

### 2.3. Molecular Identification and Phylogenetic Analysis

Eight representative *F. oxysporum* strains isolated in different years were sequenced. Mycelia were scraped from 14-day-old colonies and genomic DNA was extracted according to the instructions of the PlantGen DNA Kit CW0553A (Cwbio, Taizhou, China).

Amplification reactions were carried out in a total volume of 50 μL, containing 2 μL of template DNA, 1 μL of each forward and reverse PCR primer, 21 μL of ultrapure water, and 25 μL of 2 × PCR Taq Master Mix PC1120 (Solarbio, Beijing, China), which included 10 mM Tris-HCl (pH 8.3), 50 mM KCl, 1.5 mM MgCl_2_, 250 μM dNTP, and 0.05 U Polymerease/μL. The 18S ribosomal RNA (SSU), translation elongation factor 1-α (EF1-α), RNA polymerase II subunit 1 (rpb1), and RNA polymerase II second largest subunit (rpb2) genes were amplified with primers NS1/fung [14], EF1-728F/EF-2 [15], RPB1-Fa/RPB1-G2R and RPB2-5f2/RPB2-7cr [16], respectively. The Applied Biosystems ProFlex PCR 4484075 was used for the amplification with the following conditions: an initial denaturation step at 94 °C for 5 min, followed by 30 cycles of denaturation at 95 °C for 40 s, annealing at 47 °C for 40 s, and extension at 72 °C for 150 s, with a final extension step at 72 °C for 10 min.

PCR products were detected by 1% agarose gel in 1 × TBE buffer stained with Ethidium Bromide (Thermo Fisher 15585011, Shanghai, China) using a Nucleic Acid Electrophoresis (Thermo Fisher PSC120M, Shanghai, China), and visualized under UVP Glestudio plus (Analytik Jena, Shanghai, China), then sent to GENEWIZ, Inc. (Suzhou, China) for Sanger sequencing. Results of genome sequencing were manually edited using the software Chromas (v 2.6.6, Technelysium Pty, Ltd., Brisbane, Australia). Sequence similarity searches were analyzed by BLAST in the National Center for Biotechnology Information (NCBI) database. The obtained sequences and reference ones retrieved from NCBI GenBank were aligned by ClustalW. The maximum-likelihood tree was constructed on the three-gene (EF-1α + rpb1 + rpb2) combined datasets. Phylogenetic analysis was generated based on the general time reversible model using MEGA 7 software, and branch strength was evaluated with 1000 replicates.

### 2.4. Pathogenicity Test

Pathogenicity assays were carried out in triplicate for eight *F. oxysporum* isolates. For in vivo experiment, stems of 6-week-old healthy *P. sinese* seedlings, which had been wounded by sterile needles, were dipped in a spore suspension of approximately 10^5^ conidia/mL for 30 min, while the negative control was treated with distilled water. All plants grew in sterilized soil and were placed in a greenhouse at 24 ± 1 °C with a 12 h/12 h light and dark period. The in vitro test was conducted in the petri dish, which had been previously washed with 75% alcohol and padded with autoclaved filter paper moistened with sterilized distilled water to keep wet. Leaves were separated from *P. sinese*, wounded, and inoculated as previously described. Culture dishes were placed in the biochemical incubator. The temperature parameter was adjusted to 25 °C and the illumination condition was set to 24 h dark. Fungal pathogens were reisolated from symptomatic tissues of the infected plants and identified on the basis of cultural characteristics and molecular analysis to fulfill Koch’s postulates.

## 3. Results

### 3.1. Field Survey

All infected *P. sinese* showed typical symptoms as irregular deep reddish-brown lesions appearing on the blade tip; as the disease became worse, the whole leaf turned pale brown, then blighted, and the internal part of the stem gradually discolored. The infected plants dried out and defoliated at a severe stage. In 2018, the disease incidence was determined in the range of 54–71%, and the average among seven fields was calculated as 63%. The average disease incidence was recorded as 49%, 57%, and 52% in 2019, 2020, and 2021, respectively.

### 3.2. Morphological Identification

A total of 38 isolates of *Fusarium* were obtained from symptomatic samples. The colonies covered almost the whole petri dish in 6 days at 25 °C. Aerial mycelia were abundant, white, and floccose initially. After 3 days of inoculation, yellow–orange pigmentation developed beneath the colonies.

Microconidia were single-celled, ellipsoidal, 0–1 septate, 5–14 × 2.5–4.5 µm. Macroconidia were falcate, curved towards the ventral side, with apical cells papillate, basal cells foot-shaped, 1–4 septate (mostly 3 septate), hyaline, smooth, and thin-walled, 21–42 × 3–6 µm. Chlamydospores were globose to sub-globose with an average of 10 μm in diameter, and hyphae branched at acute angles (Figure 1). The observed morphological characteristics matched the descriptions of *F. oxysporum* [17].

### 3.3. Molecular Identification and Phylogenetic Analysis

BLAST results showed that the SSU, EF-1α, rpb1, and rpb2 sequences of eight isolates attained 97–100% identity to the corresponding sequences of *F. oxysporum* strain T1-BH.1 (MN463097), FS11719a (MN417191), GR_FOA249 (MT305083), and VPRI43194 (MN457545). Nucleotide sequences of eight strains were deposited in GenBank (Table 1).

A multilocus phylogenetic tree was constructed using the combined sequences of EF-1α, rpb1 and rpb2 (Table 2). There were a total of 1932 positions in the final dataset. The tree with the highest log likelihood (−7778.70) is displayed in Figure 2. Bootstrap values were indicated at the nodes. In the maximum-likelihood tree, eight isolates were grouped into an independent clade, supported by the 100% bootstrap value. The combined three-locus dataset revealed that eight pathogenic isolates clustered strongly (100%) with *F. oxysporum* strains. Thus, based on morphology and sequence analysis, the associated fungus was determined as *F. oxysporum*.

### 3.4. Pathogenicity Test

The pathogenicity test showed *F. oxysporum* isolates were pathogenic to *P. sinese*, exhibiting typical symptoms as observed in the field. In the in vitro assay, the detached leaves became yellow–brown and eventually blighted after 7 days of inoculation with the spore suspension. The in vivo experiment demonstrated that reddish-brown regions formed at the apex of inoculated leaves. As the disease progressed, light brown areas began to emerge in the center of the expanding red–brown lesion and spread to the entire leaf. Plants treated with pathogenic inoculums all turned wilted, and their stems revealed vascular bundle browning 25 days later. The control groups did not develop any symptoms (Figure 3). The same fungal species (*F. oxysporum*) was successfully reisolated from symptomatic tissues, fulfilling Koch’s postulates.

## 4. Discussion

*F. oxysporum* is a type of common soilborne fungi, which leads to vascular wilt and crown, stem, or root rot across a broad host range [18], including Canary Island date palm Fusarium wilt (*Phoenix canariensis* Hort. Ex Chabaud) [19], sweet basil wilt and crown rot [20], potato stem–end rot [21] and sugar beet lines root rot [22]. This fungus is also a considerable pathogen (clinically) that can infect both humans and animals [23].

Morphological observation provides clues to the pathogen’s identity as *Fusarium* spp. However, morphological features are not sufficient to distinguish many fusaria [24], and multi-locus sequence comparisons were proposed because they offer more credible information. The EF1-α sequence is extensively used to identify most *Fusarium* species [25]. In addition, combining portions of rpb1 and rpb2 is the most comprehensive phylogenetic assessment of *Fusarium* spp. as yet [6], receiving recognition for their high informativeness in analyzing *Fusarium* [26]. Thus, a phylogenetic tree was conducted consisting of concatenated partial sequences of genes for EF1-α, rpb1 and rpb2, which strongly verified that the associated pathogen was *F. oxysporum*.

When it comes to Fusarium wilt, *F. oxysporum* can colonize in the vascular bundles of the plant roots, resulting in vessel blockage, withering of the above-ground portion, along with brown necrotic spots appearing on the root epidermis [27,28,29]. Diseased crops exhibit root rot symptoms or even die. Our present work revealed that *P. sinese* was susceptible to Fusarium wilt; additionally, stem discoloration also provided evidence of vascular bundle infection.

As *Pennisetum* spp. is one of the most widely cultivated plants around the world [30], Fusarium wilt disease could cause a serious impact on forage. Located in Southwest China and the upper reaches of the Yangtze River, Chongqing enjoys a subtropical monsoon climate with four distinct seasons and abundant rainfall, leading to high pastures production. Since *P. sinese* is a primarily cultivated forage in China in the summer [31], growers of *P. sinese* should pay emphasis to *P. sinese* wilting. The identification of the causal pathogen provides a scientific basis for disease management, playing a major part in developing antifungal agents. For further research, management strategies will be investigated to minimize potential economic losses due to lower plant biomass production.

## Figures and Tables

**Figure 1 pathogens-11-00999-f001:**
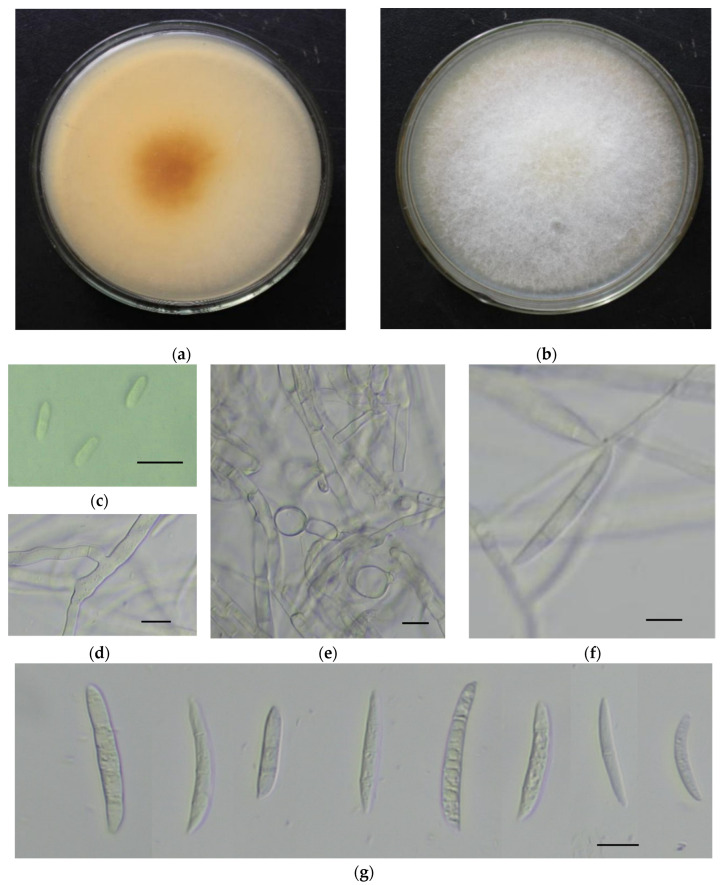
Morphological characteristics of the pathogen on PDA medium. (**a**,**b**). The back and front of the colony; (**c**) microconidia; (**d**) hyphae with acute angled branch; (**e**) thick-walled chlamydospores; (**f**) monophialides producing macroconidia; (**g**) falcate-shaped macroconidia. Scale bars in (**c**–**g**) = 10 μm.

**Figure 2 pathogens-11-00999-f002:**
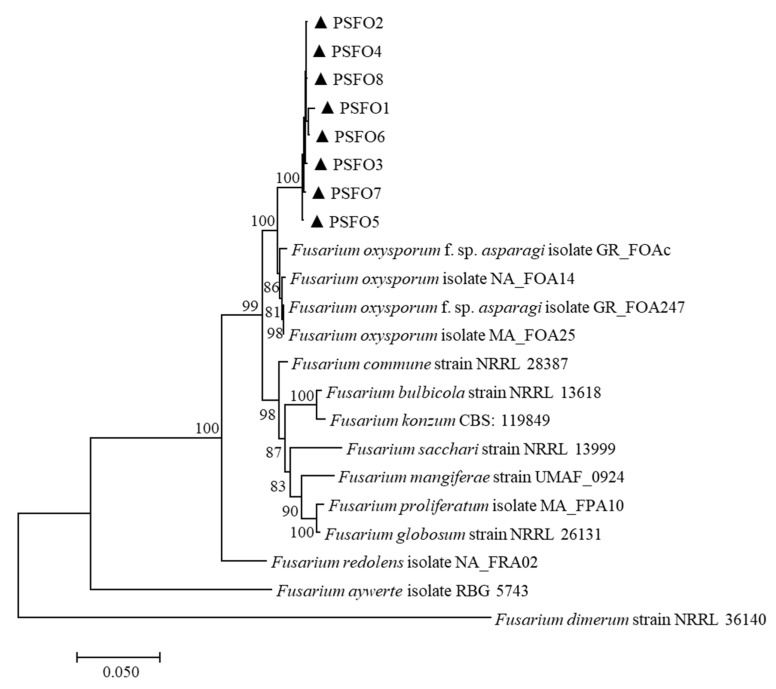
The maximum-likelihood tree was inferred from the concatenated sequences of the translation elongation factor 1-alpha (EF-1α), RNA polymerase II subunit 1 (rpb1), and RNA polymerase II second largest subunit (rpb2) genes. The scale bar represented 0.05 nucleotide substitutions per site.

**Figure 3 pathogens-11-00999-f003:**
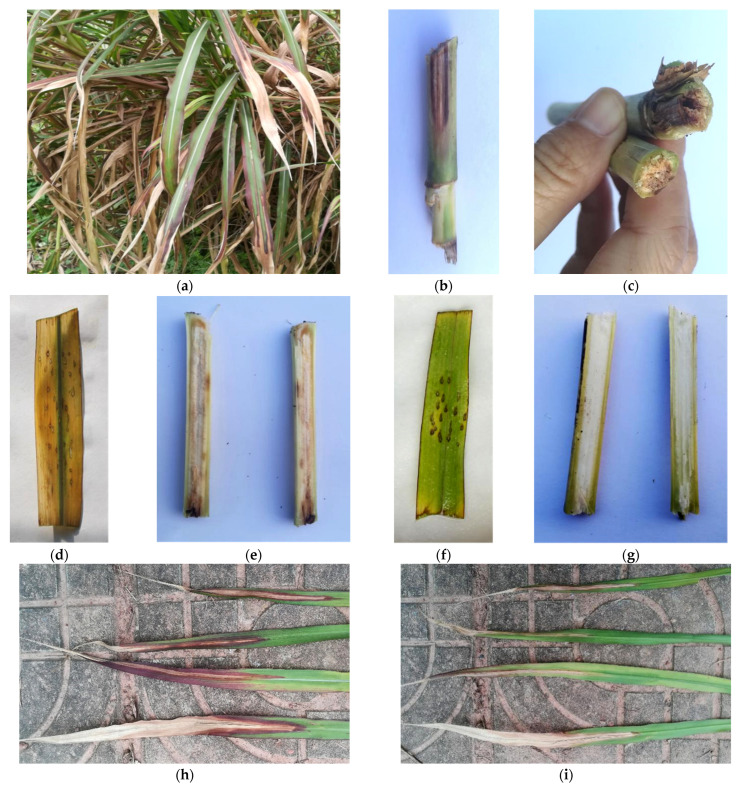
Symptoms of Fusarium wilt on *Pennisetum sinese*. (**a**) Typical symptoms on naturally infected *P. sinese* in the field; (**b**,**c**) stem discoloration on naturally infected *P. sinese*; (**d**) the detached leaf inoculated with *Fusarium oxysporum* isolate turned yellow–brown and withered; (**e**) the stem inoculated with *F. oxysporum* showed vascular bundle browning; (**f**,**g**) the control group stayed healthy; (**h**,**i**) the reddish-brown lesion appeared on leaves and infected leaves gradually wilted.

**Table 1 pathogens-11-00999-t001:** Isolates of *Fusarium oxysporum* obtained from *Pennisetum sinese* Fusarium wilt from 2018 to 2021 in China and GenBank accession numbers for DNA sequences used in this study.

Isolate	Year	GenBank Accession Number			
		SSU	EF	rpb1	rpb2
PSFO1	2018	MN954685	MN939683	MT314258	MT314259
PSFO2	2018	OP236575	OP243441	OP243434	OP243427
PSFO3	2019	OP236576	OP243442	OP243435	OP243428
PSFO4	2019	OP236577	OP243443	OP243436	OP243429
PSFO5	2020	OP236578	OP243444	OP243437	OP243430
PSFO6	2020	OP236579	OP243445	OP243438	OP243431
PSFO7	2021	OP236580	OP243446	OP243439	OP243432
PSFO8	2021	OP236581	OP243447	OP243440	OP243433

**Table 2 pathogens-11-00999-t002:** Isolates of *Fusarium* spp. and GenBank accession numbers for DNA sequences used in this study.

Fusarium Species	Isolate	GenBank Accession Number		
		EF-1α	rpb1	rpb2
*F. oxysporum*	GR_FOAc	MT305183	MT305069	MT305125
*F. oxysporum*	GR_FOA247	MT305196	MT305082	MT305138
*F. oxysporum*	NA_FOA14	MT568939	MT568955	MT568971
*F. oxysporum*	MA_FOA25	MT568947	MT568963	MT568979
*F. proliferatum*	MA_FPA10	MW091270	MW091290	MW091308
*F. redolens*	NA_FRA02	MW091278	MW091296	MW091316
*F. dimerum*	NRRL 36140	HM347133	HM347203	HM347218
*F. mangiferae*	UMAF_0924	KP753402	KP753435	KP753442
*F. aywerte*	RBG 5743	KP083250	KP083273	KP083278
*F. commune*	NRRL 28387	AF246832	JX171525	JX171638
*F. bulbicola*	NRRL 13618	KF466415	KF466394	KF466404
*F. globosum*	NRRL 26131	KF466417	KF466396	KF466406
*F. sacchari*	NRRL 13999	AF160278	JX171466	JX171580
*F. konzum*	CBS: 119849	LT996098	LT996200	LT996148

## Data Availability

The sequence data generated in this study are deposited in NCBI GenBank (https://www.ncbi.nlm.nih.gov/genbank). All accession numbers are presented in Table 1.

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
