# Peer review of "Fusarium oxysporum Associated with Fusarium Wilt on Pennisetum sinese in China"

_pathogens, 2022, doi:10.3390/pathogens11090999_

Round 1
Reviewer 1 Report (Previous Reviewer 3)
The authors revised the manuscript and it's in good shape.
L122-124: How many plants did you examine every year?
Author Response
Thanks for the reviewer’s constructive comments, which are helpful for us to revise and improve this manuscript.
In every May, seven P. sinese fields were randomly selected in the planting base, and five plots (20 m2) were investigated at each site. Approximately 70 plants per plot were included in the analysis. Thus, about 2450 plants (70×5×7) were examined every year. We have provided this information in L63-65, please check. Thanks!
Reviewer 2 Report (Previous Reviewer 2)
I think overall now the manuscript looks good. I would recommend to proof read for some minor grammer mistakes.
Author Response
Thanks for the reviewer’s constructive suggestions and kind reminders. We have carefully read through the manuscript and corrected some grammar mistakes. Thanks!
This manuscript is a resubmission of an earlier submission. The following is a list of the peer review reports and author responses from that submission.
Round 1
Reviewer 1 Report
The manuscript entitled ''Fusarium oxysporum associated with Fusarium wilt on Pennisetum sinese in China'' presents the isolation and characterization of a fungal species as the main causative agent of Fusarium wilt. I have attached the pdf version of the manuscript with major comments.
Here are few of the major suggestions:
Introduction
The introduction should be improved with more references. Please add another paragraph at the end of the introduction that generalizes the story with your major finding in the study.
Materials and methods
Please described it gently but in detail as how you perform the experiment....media composition, polymerase used etc...
Mention also the instrument or reagent used and the country of the product sourced from.
Discussion
As mentioned for the introduction, you can improve the discussion of your results by comparing similar studies.

Author Response
Thanks for the reviewer's constructive suggestions on our Communication. We have since made extensive revisions and paid particular attention to the quality of English usage with the assistance of a professional manuscript editor. Thanks.
Introduction
We have rewritten the introduction section. First, we have provided background information about pathogenic species prevalent on P. sinese. In addition, we have added another paragraph at the end of the introduction that generalizes our major findings.
Materials and methods
We are sorry for leaving out some details in the previous submission. We have added more information in our revised manuscript, and we also supplemented the instrument or reagent used and the country of the product sourced from.
Discussion
We have added a paragraph about Fusarium wilt in the discussion.
Reviewer 2 Report
This is an interesting work that provides information on the new detection of Fusarium on forage crop. I do see different approaches that was taken to identify the species that was involved to cause wilt.
However, I have few major concern regarding this manuscript.
1) Some of the symptoms that are described is not evident enough. It would be good to see detailed description of the symptoms with pictures.
2) Also there are different sections where more information is required for the readers to understand better and also to ensure that this is reproducible
3) One section on the extraction of nucleic acid from the fungus is missing and which I feel it is important. Please ensure to include this
4) Please provide detailed steps on how in-vitro test was conducted in the culture dish (line # 71)
Author Response
We appreciate the reviewer's constructive suggestions on our Communication. We are sorry that we left out some important information in the previous submission. We have made extensive modifications according to the comments. Thanks.
1) We have rewritten the description of symptoms and made it more clear.
2) We have checked through our manuscript and provided more details about our experiment.
3) We have added information on how to extract fungal nucleic acid.
Mycelia were scraped from 14-day-old colonies and genomic DNA was extracted according to the instruction of PlantGen DNA Kit CW0553A.
4) We have rewritten the pathogenicity section.
Detached leaves of 6-week-old healthy P. sinese seedlings, which had been wounded by sterile needles, were dipped in a spore suspension of approximately 105 conidia/mL for 30 min, while the negative control was treated with distilled water. The in vitro test of leaves was conducted in the petri dish, which had been previously washed by 75% alcohol and padded with autoclaved filter paper moistened with sterilized distilled water to keep wet. Culture dishes were placed in the biochemical incubator (25 °C, 24 h dark).
Reviewer 3 Report
Based on morphological and phylogenetic analysis, the authors identified Fusarium oxysporum as the pathogen causing Fusarium wilt on Pennisetum sinese. The scientific methods used in this manuscript is sound, and the results and conclusions seem reasonable. Two majors issues of this manuscript are, however, (1) sample size is too small and (2) this manuscript seems like a detailed version of First Report.
(1) Many Fusarium spp. could infect the same plant host and show the same disease symptom. It would be important to obtain enough fungal isolates and to determine that the fungus in question is indeed the only species that was associated with the disease symptom on this specific plant. Without species composition data, it is not clear if this fungus is the only species causing the disease symptom or how prevalent this fungus is among other Fusarium species.
(2) The content of this manuscript consists of sampling, identification, and Koch’s postulates. Those are the typical content for the First Report (or Research Note ) and not for a Full research paper.
Some important information is missing from the manuscript:
L35. How many plants were affected by this disease? One plant or over 50% of plants in 100 ha (how many ha)? Without disease incidence and severity information, It is not clear how important it is to study Fusarium wilt on P. sinese.
L44. How many infected tissues were sampled? How were they sampled from how many trees, from how many ha of field?
L52. Seven isolates were chosen from how many obtained isolates?
L67. …for each of seven isolates?
L68. How were stems and leaves of P. sinese seedings wounded for pathogenicity test, using what tool?
L77. Only seven isolates? Then it’s not “representative isolates (L52)”, that’s “all isolates”.
L91. Sequences from 7 isolates for each gene were identical?
L98. What do you mean by “representative sequence” that comes from one isolate?
Fig.2 Maximum likelihood is more suitable method for DNA-sequence-based phylogenetic analysis than Neighbor-Joining although the results may seem the same.
Author Response
We are grateful for the reviewer’s approval for the quality of our research results, and appreciate the reviewer's constructive suggestions on our Communication. We have since carefully revised the whole manuscript according to the comments. Thanks.
Fusarium wilt on Pennisetum sinese first appeared in Chongqing City in 2018, and our lab has observed this disease for 4 years. We obtained 38 isolates from symptomatic tissues in seven fields at first. Morphological features demonstrated that these isolates had high similarity; thus, for each field, we selected one isolate for further research. Molecular identification and pathogenicity test were conducted on all seven isolates and exhibited similar results. Therefore, we submitted sequences of one representative isolate (PFSO1) to NCBI GenBank. We have added this information in our revised manuscript.
L35. We carried out field surveys in the 400 m2 P. sinese planting base. The disease incidence of P. sinese wilting was calculated as 63% according to the formula: Disease Incidence(%) = (Number of diseased plants/Total number of plants)*100.
L44. We collected 35 infected samples from seven fields. We have provided field survey information in the revised manuscript.
L52. Seven isolates were chosen from 38 obtained isolates.
L67. Pathogenicity tests were performed with all seven isolates but exhibited similar results, so we just described isolate PFSO1 in the previous submission. We have rewritten the pathogenicity section.
L68. We wounded stems and leaves of P. sinese seedings using sterile needles. We have added this information.
L77. We felt sorry for our careless mistakes. We obtained 38 isolates at first, and these isolates were only involved in morphological characteristics, so we didn’t mention them in the previous submission. Since there was no significant difference among these isolates, we selected one isolate from each field and conducted further research. Molecular identification and pathogenicity test were conducted on seven isolates.
L91. Sorry for our inaccurate statement in the previous manuscript. BLAST results showed that the SSU, EF-1α, rpb1 and rpb2 sequences of seven isolates attained 97-100% identity to the corresponding sequences of F. oxysporum strain T1-BH.1 (MN463097), NRRL 38599 (KM092474), NRRL 34936 (JX171533) and NRRL 34936 (JX171646). Since all the isolates exhibited similar results, we just chose one representative strain and deposited its sequences in GenBank.
L98. Sorry for our careless mistakes. We mean “Sequences of one representative isolate were deposited in GenBank”. We have revised this sentence.
Fig.2 We have reconstructed the phylogenetic tree by Maximum likelihood method.
Round 2
Reviewer 1 Report
Dear Authors,
Please find attached my suggestions and comments.
You have added most of my comments in the v2.

Author Response
We gratefully appreciate the reviewer’s professional and valuable suggestions. We have carefully revised this manuscript according to the remarks. First, we have changed “cutting” into “reduce” (Line 36), and “isolates” were replaced by “F. oxysporum isolates” (Line 75). We have made Line 78-88 in one paragraph. Besides, we have added what sequences were used for the Maximum-Likelihood tree and rewritten the phylogenetic analysis section (Line 91-94). Thanks.
Reviewer 2 Report
Comments to Author:
This is an interesting work that provides information on the new detection of Fusarium on forage crop. I do see different approaches that was taken to identify the species that was involved to cause wilt.
However, I think this manuscript can be improved by validation experiments. Also there are several places, which did not address my previous comments. The line number refers to the first version of the manuscript.
1) Please provide detailed steps on how in-vitro test was conducted in the culture dish (line # 71)
2) It will be good to see the validation of the results in ITS1 and ITS4 regions.
Minor corrections:
line 35: Please mention the % of severity or incidence.- Not addressed
line 39: Figure 3 has only the foliage damage pictures occurring naturally can you please provide pictures of stem getting affected. Please also mention about the yield loss this is going to cause - Not addressed
Line 40: Please replace “Fusarium wilt” with “stem wilt” - Not addressed
Line 48: How about getting pure culture after keeping it for isolation? - Not addressed
Line 57: Please mention the recipe of the PCR mix that was used including the final concentration of the each reagents. Also please mention the brand of the each of the reagents that was used. - Not addressed
Line 82: Please provide more details of falcate shape- Not addressed
Line 86: It will be good to have a comparison table may be as supplemental data- Not addressed
Line 101: Please include more regions for comparison. - Not addressed
Line 109: Please provide age of the plant- Not addressed
Author Response
We gratefully thank the reviewer for the precious time making constructive remarks, which have greatly helped improve our manuscript. We have carefully revised this manuscript, addressing each of the comments and suggestions from the reviewer, point by point, as listed below.
1) As suggested by the reviewer, we have reorganized the pathogenicity tests section and tried to provide more details about the in-vitro assays (Line 101-106).
2) Thanks for the reviewer’s advice. ITS is the official fungal barcode; however, after reading relevant literature, we found that ITS was less effective in Fusarium identification. O’Donnell et al. have reported that ITS may not be useful in distinguishing closely related species among Fusarium [1-3]. Thus, we didn’t amplify ITS regions.
EF1-α, rpb1 and rpb2 are regarded as the most important genes for Fusarium identification, so partially sequence EF1-α, rpb1 and rpb2 genes from each selected isolate were performed for molecular analysis [4-6].
Reference:
[1] Kerry O'Donnell, Deanna A. Sutton, Michael G. Rinaldi, et al. 2010. Internet-accessible DNA sequence database for identifying fusaria from human and animal infections. Journal of Clinical Microbiology. 48, 3708-3718.
[2] Kerry O'Donnell, Elizabeth Cigelnik, Helgard I. Nirenberg. 1998. Molecular systematics and phylogeography of the Gibberella fujikuroi species complex. Mycologia 90, 465-493.
[3] Kerry O'Donnell, Elizabeth Cigelnik. 1997. Two divergent intragenomic rDNA ITS2 types within a monophyletic lineage of the fungus Fusarium are nonorthologous. Molecular Phylogenetics and Evolution 7, 103-116.
[4] Abd Rahim Huda-Shakirah, Masratul Hawa Mohd. 2021. First report of Fusarium sacchari causing leaf blotch of orchid (Dendrobium antennatum) in Malaysia. Crop Protection 143: 105559.
[5] Kerry O’Donnell, Alejandro P. Rooney, Robert H. Proctor, et.al. 2013. Phylogenetic analyses of RPB1 and RPB2 support a middle Cretaceous origin for a clade comprising all agriculturally and medically important fusaria. Fungal Genetics and Biology 52, 20-31.
[6] M. H. Laurence, J. L. Walsh, L. A. Shuttleworth, et.al. 2016. Six novel species of Fusarium from natural ecosystems in Australia. Fungal Diversity 77, 349-366.
Minor corrections:
Line 35: We thank the reviewer for pointing out this issue. We have added field study information and mentioned the % of severity or incidence. The disease incidence was calculated as 63% (Line 115-116). Please check “2.1. Disease incidence and sampling collection” and “3.1. Field survey”.
Line 39 & Figure 3: Thanks for your suggestion. We have provided pictures of stem getting affected in the revised manuscript (Figure 3b&3c). In addition, we have mentioned that Fusarium wilting diseases may have potential dangers to forage yield (Line 49).
Fusarium wilt on Pennisetum sinese first appeared in Chongqing City in 2018, and our lab has observed this disease for 4 years. We have not yet systematically conducted yield loss research, and this study will be carried out in the near future. Our preliminary experiment showed that the forage output of a healthy P. sinese field was about 20 tons. After being infected by Fusarium oxysporum, the field production decreased by 1/3 to 1/2.
Line 40: We have replaced “Fusarium wilt” with “stem wilt”.
Line 48: We are sorry for missing this important information. Pure cultures of each isolate were obtained by subcultivating hyphal tips on PDA. We have added this sentence in the revised manuscript (Line 70-71).
Line 57: Thank you for your reminding. We feel sorry for our careless mistakes. We used 2×PCR Taq Master Mix PC1120 (Solarbio, Beijing, China). We have added the brand of the reagents that were used in our experiment (Line 80).
The final concentration of PCR Master Mix PC1120 contained 10 mM Tris-HCl (pH 8.3), 50 mM KCl, 1.5 mM MgCl2, 250 μM dNTP each and 0.05 U Polymerease/μl.
Line 82: Macroconidia were falcate, curved towards the ventral side, with apical cells papillate, basal cells foot-shaped. We have revised and described the falcate shape in more detail (Line 123-124).
Line 86: We have supplemented a table, which provides GenBank accession numbers for DNA sequences used in this study. Please check “Table 1”.
Line 101: Thanks for the reviewer’s comments. EF1-α has great advantage for Fusarium identification because it can frequently resolve at the species level. Additionally, the sampling of the EF1-α region of Fusarium genus was abundant [4]. Rpb1 and rpb2 are noted for their informativeness in analyses of Fusarium species since they can be aligned easily across the breadth of the genus and beyond [5-6]. EF1-α, rpb1 and rpb2 are regarded as the most important regions for distinguishing Fusarium. Hence, identifying Fusarium species based on the regions of EF1-α, rpb1 and rpb2 seems to be sufficient.
Reference:
[4] Abd Rahim Huda-Shakirah, Masratul Hawa Mohd. 2021. First report of Fusarium sacchari causing leaf blotch of orchid (Dendrobium antennatum) in Malaysia. Crop Protection 143: 105559.
[5] Kerry O’Donnell, Alejandro P. Rooney, Robert H. Proctor, et.al. 2013. Phylogenetic analyses of RPB1 and RPB2 support a middle Cretaceous origin for a clade comprising all agriculturally and medically important fusaria. Fungal Genetics and Biology 52, 20-31.
[6] M. H. Laurence, J. L. Walsh, L. A. Shuttleworth, et.al. 2016. Six novel species of Fusarium from natural ecosystems in Australia. Fungal Diversity 77, 349-366.
Line 109: We used 6-week-old healthy P. sinese seedlings for pathogenicity tests, and we have mentioned the age of plants in “2.4. Pathogenicity test” section (Line 97).
We sincerely appreciate the reviewer’s suggestions. If there are any other modifications we could make, we would like very much to revise them. Once again, thanks for the reviewer’s careful checks and professional comments.
Reviewer 3 Report
The manuscript was revised and improved. The authors' answers to the reviewers' questions seem satisfactory and they also added necessary information in the revised manuscript. I would recommend the publication of this revised manuscript to the Journal of Pathogens.
Minor comments and suggestions:
L107: ...in the range of 54-71% .. How many plants did you examine? Or mention about how many plants are there in each field/plot?
L133, L136: Table 1(?)
L136: Isolates of Fusarium spp. and GenBank accession numbers for DNA sequences used in this study.
Author Response
We sincerely thank the reviewer for providing very helpful comments to guide our revision. We have carefully revised our manuscript according to the reviewer’s suggestions. Thanks.
L107: Approximately 70 plants per plot were included in the analysis. We have added this information in the revised manuscript (Line 114-115).
L133, L136: We have changed “Table” into “Table 1” (Line 143, Line 147).
L136: Thanks for your suggestion. We have inserted this sentence (Line 147).